**Data Availability Statement:** All relevant data are within the paper.

**Funding:** This research was funded by the CNADRI Technology R&D Program, grant number 2021-

# Creep mechanical tests and shear rheological model of the anchorage rock mass under water–rock coupling

Zhao Zhongliang[1], Dai Wukui[1], Yang Jianjun🟢[1]*, Zhou Mi[2], Liu Ziwei[1]

1 China Northeast Architectural Design & Research Institute Co, Ltd, Shenyang, Liaoning, China, 2 ZJDS Geotechnical Engineering CO., LTD, Shenyang, Liaoning, China

* jianjunyang147@163.com

## Abstract

The development of deep geotechnical engineering is restricted by the complex geological conditions of deep rock masses and the unknown creep mechanism of rock in water-rich environments. To study the shear creep deformation law of the anchoring rock mass under different water content conditions, marble was used as the bedrock to make anchoring specimens, and shear creep tests of the anchoring rock mass under different water contents were carried out. The influence of water content on rock rheological characteristics is explored by analysing the related mechanical properties of the anchorage rock mass. The coupling model of the anchorage rock mass can be obtained by connecting the nonlinear rheological element and the coupling model of the anchorage rock mass in series. Related studies show that (1) shear creep curves of anchorage rock masses under different water contents have typical creep characteristics, including decay, stability and acceleration stages. The creep deformation of the specimens can be improved with increasing moisture content. (2) The long-term strength of the anchorage rock mass shows an opposite change law with increasing water content. The creep rate of the curve increases gradually with increasing water content. The creep rate curve shows a U-shaped change under high stress. (3) The nonlinear rheological element can explain the creep deformation law of rock in the acceleration stage. By connecting the nonlinear rheological element with the coupled model of anchoring rock mass in series, the coupled model of water–rock under water cut conditions can be obtained. The model can be used to study and analyse the whole process of shear creep of an anchored rock mass under different water contents. This study can provide theoretical support for the stability analysis of anchor support tunnel engineering under water cut conditions.

## 1 Introduction

Different from those encountered during shallow geotechnical engineering, the geological environments of deep rocks are extremely complex and can be summarized by a high temperature, high stress, high permeability and underground water, among which a large underground

DBY-KY-06 and DBY-QN-2022-05; Cscec Technology R&D Project, grant number CSCEC-2020-Z-57.

**Competing interests:** The authors have declared that no competing interests exist.

water is one of the most common triggers of geological disasters. The study of creep mechanical tests and shear rheological model of the anchorage rock mass under water–rock coupling will facilitate a better understanding of the long-term stability of rock masses. The research direction of tunnel excavation and rock mass engineering is gradually transforming from shallow to deep, and rock engineering will be faced with complex engineering geological conditions [1, 2]. During tunnel excavation, anchorage support can improve the mechanical properties of the rock mass. However, in actual geotechnical engineering, the excavation of the surrounding rock and tunnel is often below the groundwater level, which leads to shear creep deformation and failure of the anchored rock mass under the long-term softening effect of groundwater [3, 4]. The change in groundwater level will affect the anchoring effect of the surrounding rock tunnel, thus reducing the stability of rock mass engineering [5–7].

The rheological characteristics of the surrounding rock under long-term loading will affect the stability of the rock. To explore the creep characteristics and deformation rules of rock salt under uniaxial compression, Singh [8] used a uniaxial compression testing machine to predict the creep behaviour of rock salt and studied the failure characteristics of rock salt using acoustic emission technology. Mansouri [9] proposed a constitutive model to describe the rheological characteristics of rock based on the uniaxial compression test of rock salt and used the test data to illustrate the accuracy of the model. Liu et al. [10] used the single specimen method to measure the mechanical parameters of mudstone in the creep process, established the coupling function relationship between the mechanical parameters of mudstone with the stress level, long-term strength and time, and obtained the general expression of the damage law of the mechanical parameters of mudstone. Shlyannikov [11] used mechanical tests to explore the creep damage characteristics of rock in plane and three-dimensional space and proposed a stress intensity factor to evaluate the degree of rock failure. Jv et al. [12] conducted a triaxial compression creep test on red-bed mudstone and established an improved Burgers model to explore the nonlinear characteristics of mudstone. Zhao et al. [13] showed that the higher the bolt density, the better the control effect of rock rheology, and there was a reasonable matching interval between the bolt support density and rock rheological parameters through the indoor creep test and the establishment of the rheological constitutive equation of the anchor body. The above studies mainly studied the creep characteristics of rock by using different test methods, but the related studies did not consider the complex environment of the surrounding rock.

Relevant scholars have studied the mechanical properties and failure mechanism of rocks under water-bearing conditions. Hossein [14] carried out mechanical tests of sandstone under different water-bearing conditions and established an empirical formula for the influence of water content on the mechanical behaviour of sandstone. Vergara [15] carried out unconfined expansion tests and static pressure expansion tests to determine the expansion behaviour of rocks and used the water content index to evaluate the mechanical properties of muddy expansive rocks. Lv et al. [16] obtained the creep deformation rule and failure mechanism of sandstone under complex geological conditions by conducting fluid-solid interaction creep tests on saturated rock masses. Fu et al. [17] established a mechanical analysis model of a water-resistant rock beam of a floor according to the characteristics of a pressurized floor and studied the variation trend of deflection and internal stress of an effectively water-isolated rock beam under the combined action of mining stress and water pressure by using the principle of virtual work and the variational condition of the energy functional. Antonio [18] studied the deep tunnel of elastic rock with anisotropy and permeability and gave the closed solution of surrounding rock stress and deformation under the action of groundwater seepage. According to the creep characteristics of water-bearing soft rock, Yu et al. [19] proposed a nonlinear shear-accelerated creep starting model reflecting the influence of water content to describe the

whole creep process of water-bearing soft rock and verified the model through experimental results. The above studies mainly focus on the failure modes of different rocks under water-bearing conditions, but there are relatively few studies on the creep characteristics of anchorage rock masses.

In this paper, the rheological deformation of the surrounding rock in the tunnel support below the groundwater level is taken as the research background. First, shear creep tests of anchoring rock masses under different water contents are carried out. The strength relation is introduced into the Weibull distribution and connected in series with the constitutive model of the anchorage rock mass; thus, the constitutive model describing the whole rheological process of the anchorage rock mass under the interaction of water and rock is obtained. In this paper, the original water-rock coupling model was optimized by exploring the quantitative relationship between water content and shear modulus, and nonlinear rheological elements are introduced to characterize the deformation characteristics of the anchorage rock mass at the acceleration stage. Finally, the advantages of the coupling model are illustrated by comparison with other traditional models.

## 2 Test plan

### 2.1 Sample preparation

In the shear creep test of the anchorage rock mass under water cut conditions, marble from a certain place in Haizhou open-pit mine, Fuxin City, Liaoning Province, was used as the test specimen, and the overall size of the test specimen was a cube of 100 mm×100 mm×100 mm [20]. First, a rock cutter was used to cut the whole rock block to reduce the discreteness of the test. The cutting size was 50 mm×100 mm×100 mm, and a hole with a diameter of 10 mm was drilled at the centre of the rock block by a rock drilling machine. Then, cement mortar is used to bond the upper and lower walls of the rock block. The mixing ratio of cement mortar is cement: river sand: water = 1:1.5:0.8, which meets the relevant test requirements [21]. The joint thickness is 5 mm, the joint shape, size and material ratio are consistent in the test process, and the indoor curing lasts for 28 days. Finally, a steel bar with a diameter of 6 mm is installed into the drilling hole of the rock block, and the overall length of the bolt is 100 mm. The material of the bolt is HRB335 steel, and the yield strength is 335 MPa. The grouting material was used to fill the holes and pores, the grouting material was consistent with the joint material, and the indoor curing lasted 28 days. The specimen is shown in Fig 1. To obtain the conventional mechanical parameters of the anchorage rock mass, uniaxial rock tests and shear tests were carried out on the anchorage rock mass in the laboratory, and the test results are shown in Table 1. A uniaxial compression test was used to determine the uniaxial compressive strength (UCS), Elastic modulus and Poisson's ratio of the specimen, while shear strength, cohesion and internal friction angle were obtained by a direct shear test (Fig 2). For the mechanical parameters of the specimen, the UCS was tested in the direction parallel to the joint, the direct shear test was conducted along the joint. The results are shown in Fig 2.

The saturated water content of the rock is 0.16% [22, 23]. In this test, the water-immersion device of the true triaxial testing machine was used to treat the specimen with water, as shown in Fig 3(A). The water content of the samples was controlled by the soaking time. The corresponding test results are shown in Fig 3(B). The samples were divided into four groups: A (dry, moisture content 0), B (soaked in water for 6 h, average water content 0.08%), C (soaked in water for 12 h, average water content 0.13%), and D (soaked in water for 24 h, saturated state, average water content 0.16%).

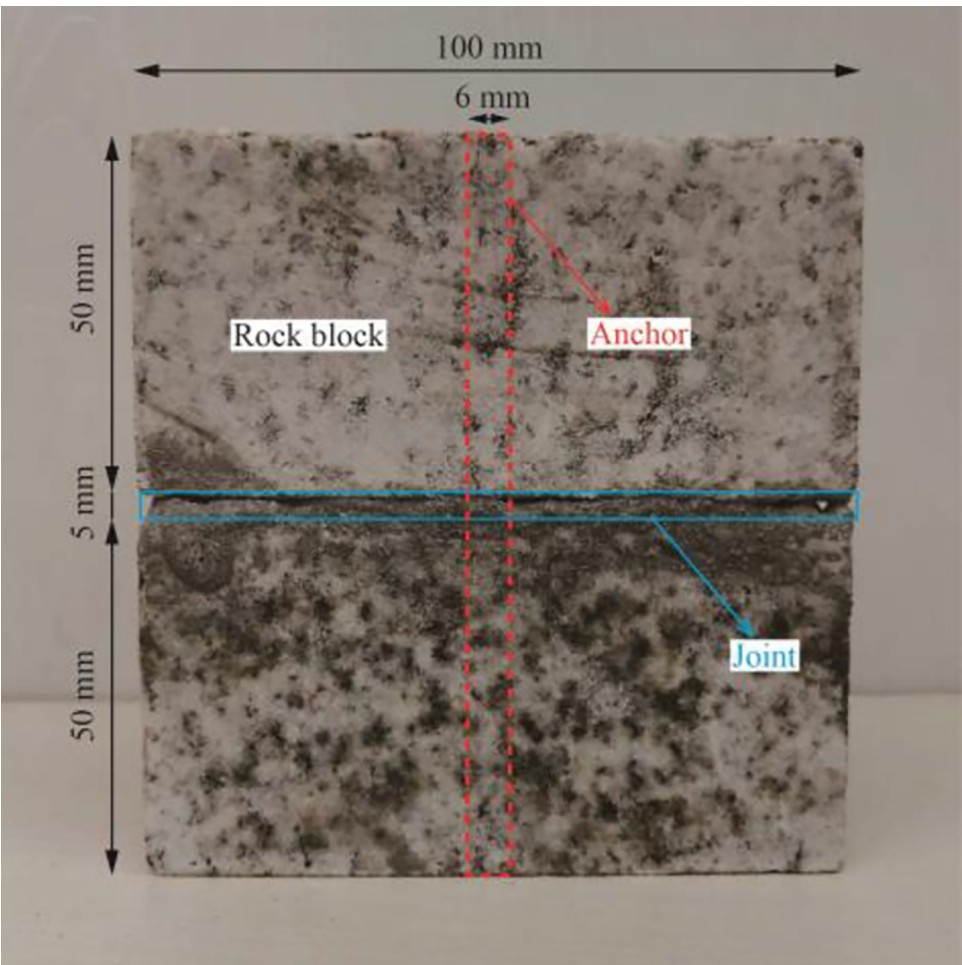

**Fig 1. Schematic diagram of the anchorage rock mass.**

## 2.2 Test equipment and steps

A Taw-2000 microcomputer was used to control the shear test device in the electrohydraulic servo rock triaxial test instrument, as shown in Fig 4. Its maximum test force was 2000 kN. Axial displacement was used to control the shear test, and the loading speed was kept at 0.1 mm/min. The TAW2000 testing machine is composed of a loading system, measuring system, controller and other parts. It adopts microcomputer-controlled electrohydraulic servo valve loading and manual hydraulic loading to complete automatic control. The testing machine have the ability to automatically complete the rocks' uniaxial and triaxial compression tests, uniaxial & triaxial rheological tests, and shear composite tests. In the test, the electro-hydraulic servo proportional valve group with wide range of speed regulation and computer digital control is applied to automatically and accurately realize the tests of the axial and radial constant

**Table 1. Mechanical parameters of the specimens.**

| Parameter | Compressive strength/MPa | Shear strength/MPa | Elastic modulus/GPa | Poisson's ratio | Cohesion/MPa | Internal friction angle (°) |
|---|---|---|---|---|---|---|
| Average value | 118.2 | 10 | 20.7 | 0.25 | 6.9 | 35.2 |

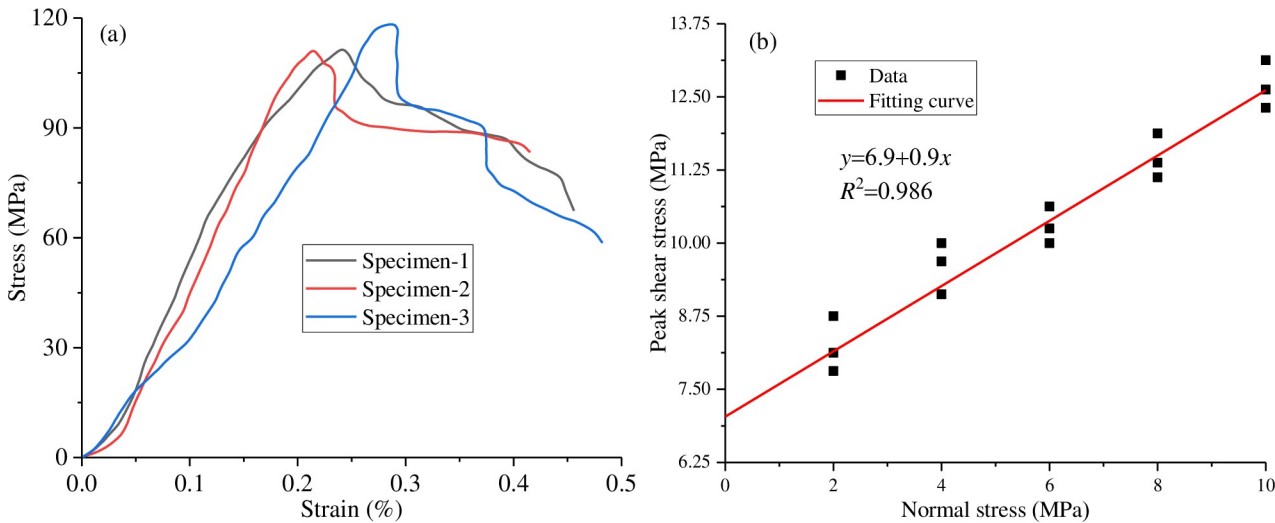

**Fig 2.** Conventional curve: (a) Uniaxial stress-strain curve; (b) Peak shear stress curve under normal stress.

stress, the constant strain, and the constant displacement. It can dynamically display the whole process of the test.

Shear creep tests of the anchorage rock mass under different water contents are carried out by stepwise loading [24]. The shear strength of the anchorage rock mass is obtained through conventional mechanical tests to determine the shear stresses at all levels of the test. The test grades were divided into four levels of loading, and the shear stress levels of each level were 20%, 40%, 60% and 80% of the shear strength of the specimen. The specific steps of the shear creep test of the anchorage rock mass under different water contents were as follows:

1. Apply normal stress: the anchorage rock mass under different water contents is put into the shear device, and the normal load is applied to the predetermined target at a rate of 20 kPa/s (the normal stress is set as 4 MPa, which is obtained by selecting 40% of the shear strength of the specimen) and remains unchanged.

2. Shear stress is applied: after the normal stress is stable, horizontal shear load is applied step by step at a rate of 20 kPa/s from low to high, and instantaneous displacement is measured and read immediately after each level of shear stress is applied.

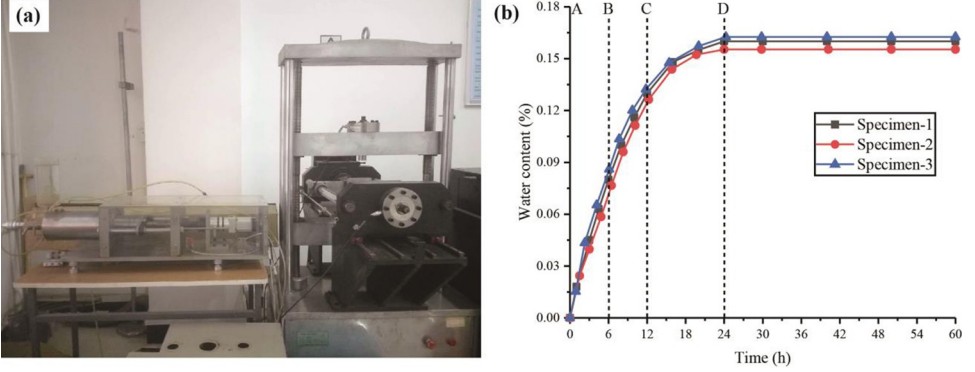

**Fig 3.** Water content test: (a) immersion device; (b) test results.

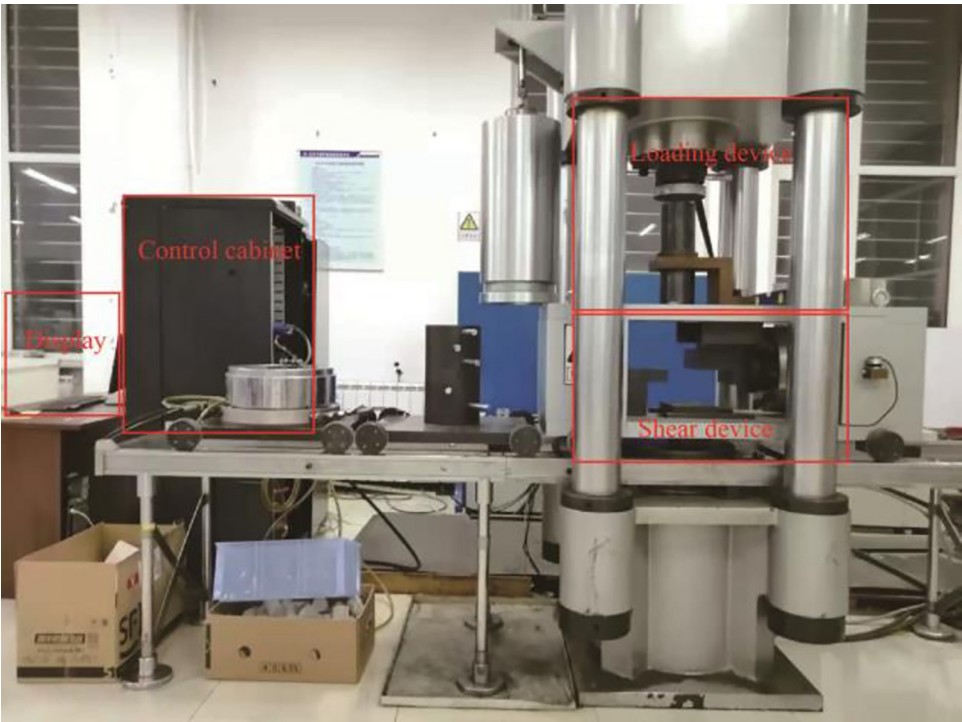

**Fig 4. Loading device diagram for the shear test.**

3. Data reading: The shear strain values of specimens are read at intervals of 5 min, 10 min, 20 min, 40 min, 1 h, 2 h, 4 h, 8 h and 12 h and then read every 12 h. The stability condition of rock mass shear deformation is that the strain rate increment is $\leq 5 \times 10^{-4}$ mm/d [25]. The specific test scheme is shown in Table 2.

## 3 Results and analysis

### 3.1 Test result

By carrying out graded loading shear creep tests on the anchorage rock mass under different water contents, the deformation curves of the specimens under different stress levels are obtained. The test results are shown in Fig 5. It can be seen from the test curve that the creep characteristics under different stress levels are different. The creep deformation under high stress includes decay, steady and acceleration stages, while the creep deformation under low stress only includes decay and stability stages.

**Table 2. Test scheme.**

| Test group | Water content/% | Shear stress/MPa | Shear strength/MPa |
|---|---|---|---|
| Group A | 0 | 2-4-6-8 | 10 |
| Group B | 0.08 | 2-4-6-8 | 10 |
| Group C | 0.13 | 2-4-6-8 | 10 |
| Group D | 0.16 | 2-4-6-8 | 10 |

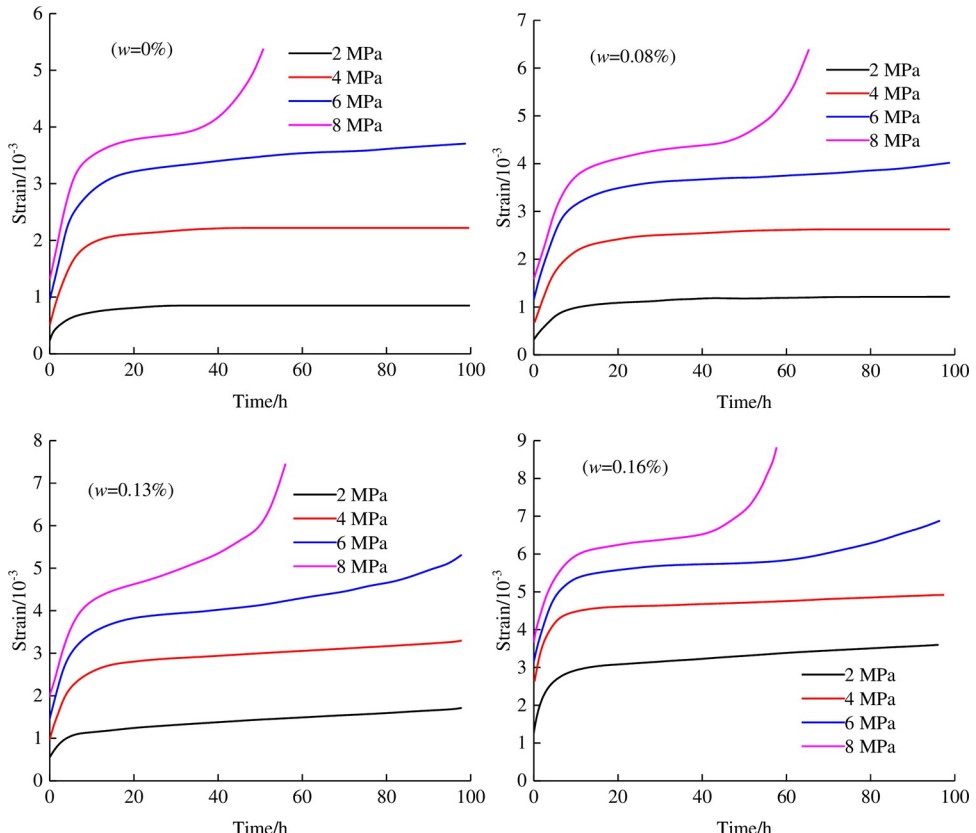

**Fig 5. Shear creep curves of the anchorage rock mass under different water contents.**

Fig 5 shows that the shear creep test curves of the anchorage rock mass under different water contents have typical creep characteristics, which can be divided into three stages according to the characteristics of the test curves:

1. Decaying creep stage: with the increase in shear load, the anchorage rock mass first reaches the instantaneous creep deformation and then directly enters the decaying creep stage. The application of the shear stress level leads to the gradual closure of the original pores in the specimen, and the continuous compaction of the pores in the specimen leads to the gradual increase in the creep deformation of the specimen. However, the creep deformation rate shows a gradual downwards trend and tends to zero with increasing creep time.

2. Steady creep stage: with the continuous application of shear stress, new cracks and pores appear in the specimen, and the creep rate of the specimen remains approximately constant and approximately 0. The reason is that the tightness of the original pores inside the specimen increases the stiffness of the specimen, which leads to the improvement of the ability of the specimen to resist shear deformation. There are two scenarios in this phase: Under the condition of low stress, the creep deformation is constant, and the creep rate is 0, which is called the stable creep stage. Under the condition of high stress, the gradual increase in the shear stress level leads to the appearance of constant shear stress in the middle and late stages of high stress creep, but the linear change in the shear strain of the specimen increases gradually; that is, the creep rate is greater than 0, which is called the unstable creep stage.

3. Accelerated creep stage: with the increase in the shear stress level, the newly formed pores of the specimen at this stage gradually expand into cracks and gradually evolve from mesoscopically to macroscopically until the shear stress level reaches the shear strength of the specimen. When macroscopic cracks appear in the specimen, the creep variables show a nonlinear increasing trend, and the corresponding creep rate also increases rapidly.

The deformation of specimens at the steady-state stage is different under different water contents, mainly manifested as the increase in water content leading to the greater deformation of specimens. The reasons are as follows: With increasing water content, the interlayer dislocation of pores becomes easier, which reduces the compressive strength of the specimens and increases the ductility of the specimens. When the water content is constant, the deformation of the specimen increases with increasing shear stress. The larger the water content is, the larger the deformation corresponding to the steady state stage. However, it can be seen from the figure that the increase in deformation gradually decreases, indicating that the ability of the specimen to resist compression deformation gradually increases in the steady state stage.

## 3.2 Long-term strength analysis

Through the above analysis, it can be seen that the shear strength and deformation and failure of rocks under different water contents have different changing trends. To explore the change law of the shear strength of the anchorage rock mass under different water contents, the Boltzmann superposition principle was used to draw the isotemporal curve of the shear stress–displacement of the anchorage rock mass under different water contents. The isochronal stress–strain curve is shown in Fig 6. As can be seen from the figure, the curve cluster gradually diverges from aggregation, marking the transformation of the specimen from viscoelastic deformation stage to viscoplastic deformation stage, indicating that there is an obvious divergence starting point of the curve.

The long-term strength of the anchorage rock mass is related to the water content. With increasing water content, the long-term strength of the corresponding specimen shows a gradual decline (Fig 7). The reason is that the increase in water content plays a role in weakening the shear strength of the rock to a certain extent, leading to the gradual decline in the shear strength of the specimen. The curve is approximately linear before the long-term strength (The curve below the dotted line). With the increase in shear stress, the shear displacement gradually increases, and the crack of the specimen experiences a process from closure to gradual expansion. The reason is that with the increase in the shear stress level, the ability of the rock mass to resist deformation decreases. When the shear strength is exceeded, the specimen has been damaged, and the rock mass has lost the ability to resist shear deformation.

## 3.3 Creep rate analysis

To better study the shear creep process of rock under different shear stress levels and water contents, the change curve of the shear creep rate of the anchored rock mass under different water contents with time is drawn, and the test curve is shown in Fig 8.

As a whole, the creep rate under different water contents shows a trend that decreases sharply in a short period of time and then approaches a certain value gradually. When the time is fixed, the increase in the shear stress level will lead to an increase in the creep rate. The reason is that the shear creep rate describes the creep variable per unit time. The larger the shear stress is, the larger the shear displacement will be under the same normal stress and water content. At the same time, when the shear stress level is fixed, the corresponding creep rate increases gradually with increasing water content, and the maximum creep rate shows a linear

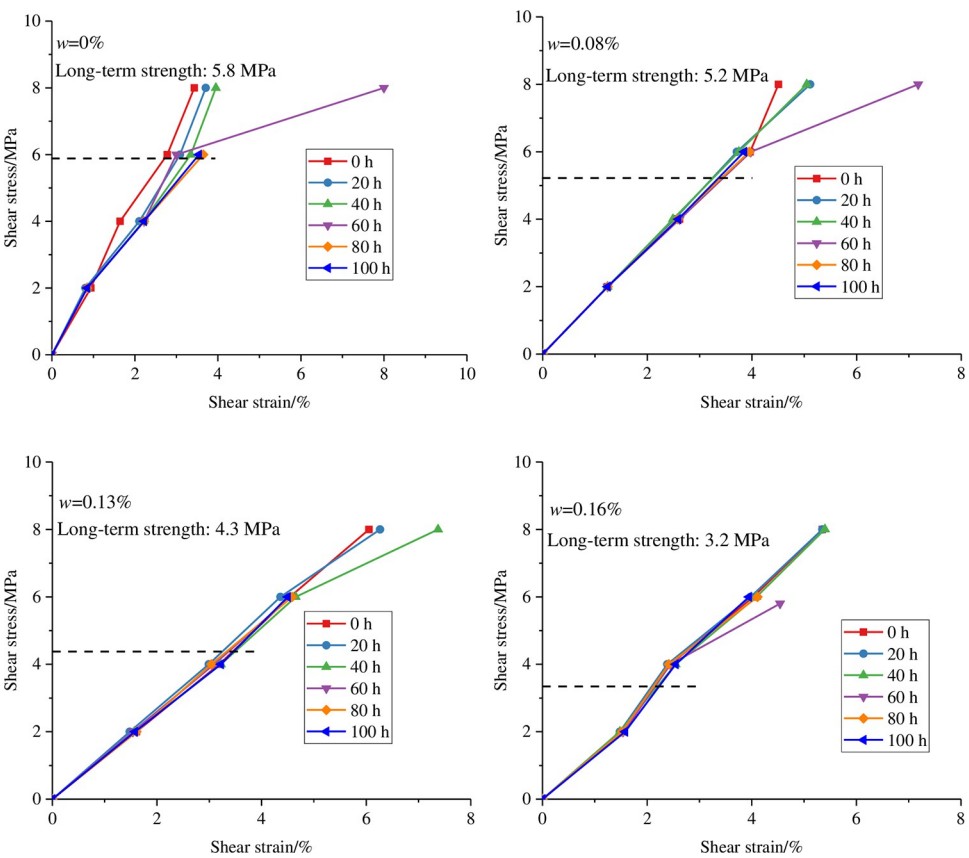

**Fig 6. Isochronal stress–strain curves of the anchorage rock mass under different water contents.**

expansion trend with increasing water content. This indicates that the expansion of the water content promotes the shear creep deformation of rock. In the decaying creep stage, the creep rate shows a cliff-like decline with time. The reasons are as follows: in the early stage of shear creep, the creep rate decreases greatly because the rock has the ability to resist shear failure. In the steady state or metastable creep stage, the creep rate curve decreases slowly and gradually approaches 0. At this stage, the joint cracks develop and expand gradually under the action of fracture water, resulting in the gradual decline in the resistance to shear failure of the specimen. In the accelerated creep stage, crack penetration occurs in the specimen, the creep rate increases, and the curve shows a U-shaped change under high stress levels.

## 4 Influence of water content on rock rheology

The results show that the shear creep curve of the anchorage rock mass can be roughly divided into the decay stage (the specimen is in the state of compaction), steady stage (the specimen is in the state of crack expansion) and acceleration stage (the specimen is in the state of crack penetration). The influence of different water contents on the shear rheology of the anchoring rock mass is mainly reflected in creep deformation caused by shear failure caused by interlayer dislocation, which affects the acceleration stage of shear failure of the rock mass and finally results in specimen failure. In this paper, the influence of water content on the shear strength of anchorage rock during shear creep is considered, and then the influence law of water content on the creep fracture of specimens is explored.

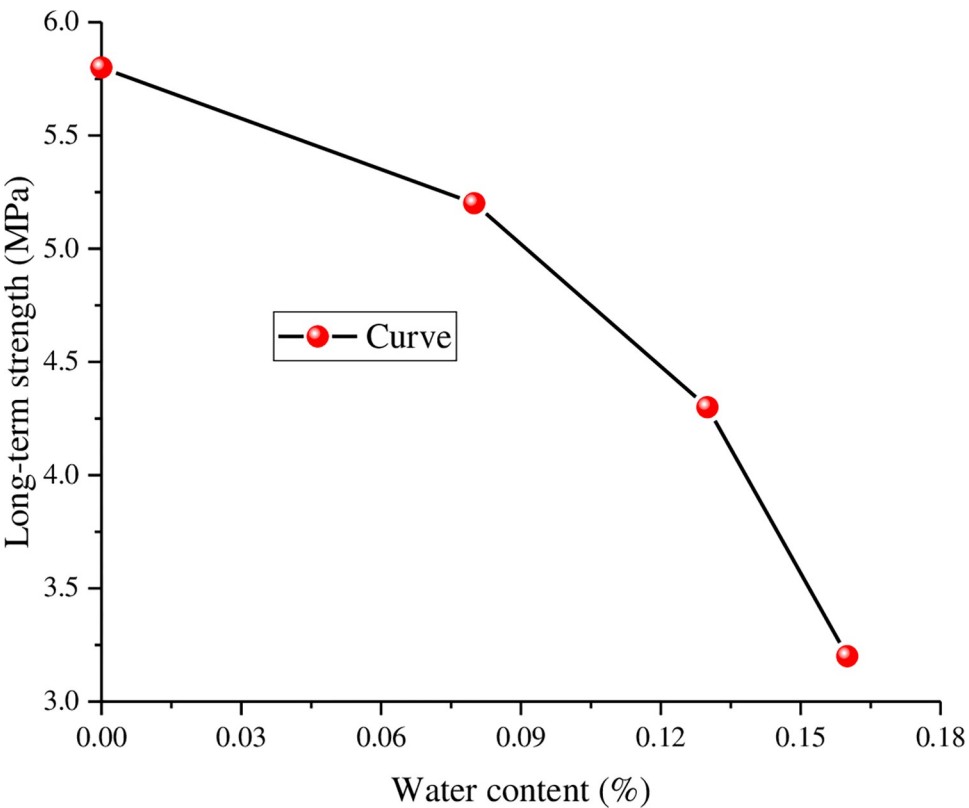

**Fig 7. Curve of long-term strength with water content.**

As an inherent property of rock, the shear modulus of an anchorage rock mass can well reflect the strength characteristics of rock. Therefore, the average shear modulus of the anchorage rock mass is selected as the research object [26]. The change curves of the shear modulus of specimens under different water contents were drawn by calculation, as shown in Fig 9. With the gradual increase in water content, the shear modulus of the specimen shows nonlinear changes. Through fitting, it can be seen that the change trend of the shear modulus and water content conforms to an exponential change, so the change function of the shear modulus with water content is as follows:

$$G(\omega) = a \times \exp(b \times \omega) + c \tag{1}$$

where $a$, $b$ and $c$ are function coefficients; $G(\omega)$ is the calculated average value of the shear modulus of the anchored rock mass; and $\omega$ is the water content.

## 5 Study on constitutive model

### 5.1 Constitutive model of anchored rock mass

The rock mass constitutive model is often simulated by the General Kelvin model, which is composed of a spring and Kelvin model in series, and its constitutive equation is [27, 28]:

$$\frac{\eta_k}{E_h + E_k}\dot{\sigma}_k + \sigma_k = \frac{E_h E_k}{E_h + E_k}\varepsilon_k + \frac{\eta_k E_h}{E_h + E_k}\dot{\varepsilon}_k \tag{2}$$

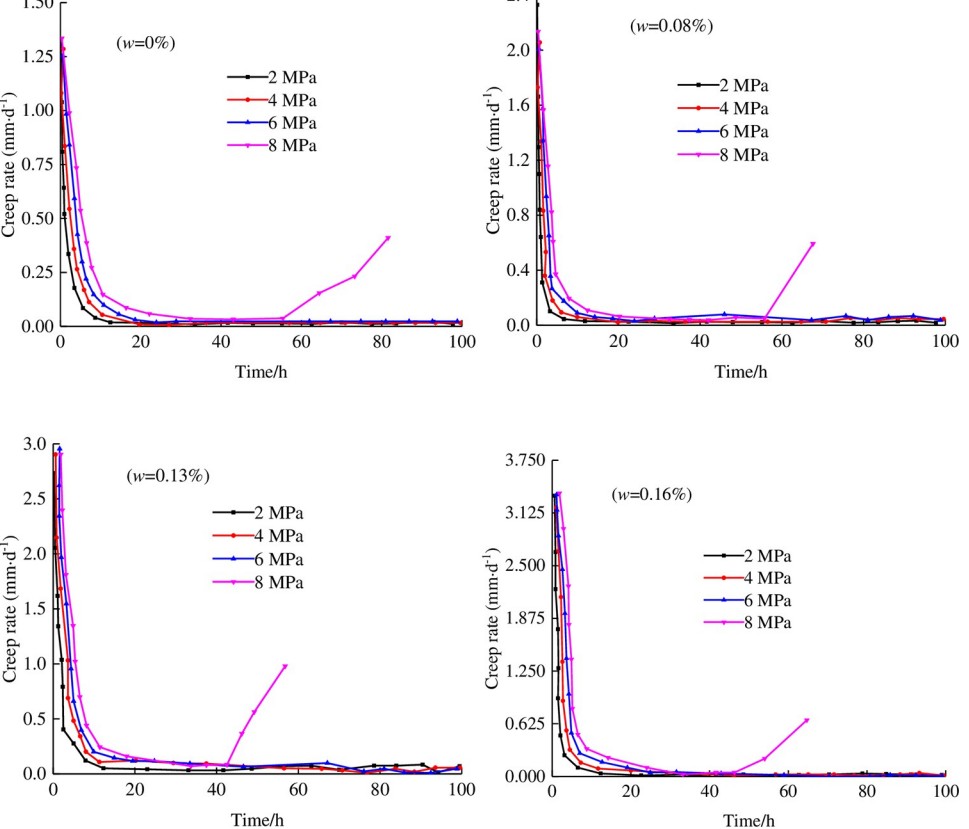

**Fig 8. Shear creep rating-time curves of the anchorage rock mass under different water contents.**

where $\sigma_k$ is the rock mass stress, $\varepsilon_k$ is the rock strain, $E_h$ is the instantaneous elastic modulus, and $E_k$ is the hysteresis elastic modulus. $\eta_k$ is the viscosity coefficient.

The rock bolt loss in the rock mass is mainly caused by shear creep of the rock mass and rock bolt deformation. The rock bolt will exhibit deformation and bending under the action of rock. However, the bending deformation of the rock bolt will lead to the gradual decline of the rock bearing capacity, so the coupling effect between rock mass creep and rock bolt loss occurs in the anchorage rock mass. In this paper, the elastic element describing the bolt is connected in parallel with the generalized Kelvin body describing the rock mass to characterize the cooperative deformation of the anchorage rock mass. The corresponding model is shown in Fig 10.

In the figure, $E_s$ is the elastic modulus of the bolt, and the other meanings are described above. The model is connected in parallel [29, 30], and its stress–strain law is shown in Eq (3):

$$\begin{cases} \varepsilon = \varepsilon_k = \varepsilon_s \\ \sigma = \sigma_k + \sigma_s \end{cases} \tag{3}$$

where σ and ε are the coupled stress and strain of the anchorage rock mass, respectively. In the subscript, k represents the rock mass, and s represents the rock bolt.

The constitutive equation of the elastic model of the rock bolt is:

$$\sigma_s = E_s \varepsilon_s \tag{4}$$

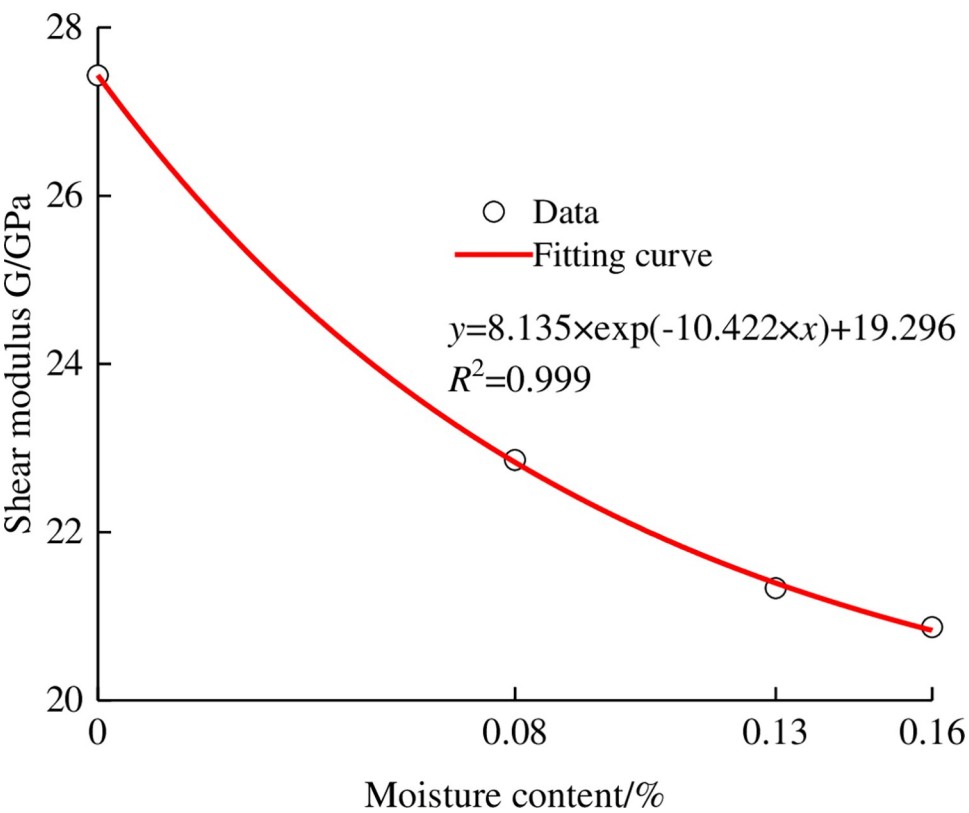

**Fig 9. Change curve of the shear modulus of the anchorage rock mass under different water contents.**

Then, the constitutive equation of the coupled model of the anchorage rock mass is:

$$\sigma + \frac{\eta_{\mathrm{k}}}{E_{\mathrm{h}} + E_{\mathrm{k}}}\dot{\sigma} = \left(\frac{E_{\mathrm{h}}E_{\mathrm{k}} + E_{\mathrm{h}}E_{\mathrm{s}} + E_{\mathrm{s}}E_{\mathrm{k}}}{E_{\mathrm{h}} + E_{\mathrm{k}}}\right)\varepsilon + \frac{\eta_{\mathrm{k}}(E_{\mathrm{h}} + E_{\mathrm{s}})}{E_{\mathrm{h}} + E_{\mathrm{k}}}\dot{\varepsilon} \qquad (5)$$

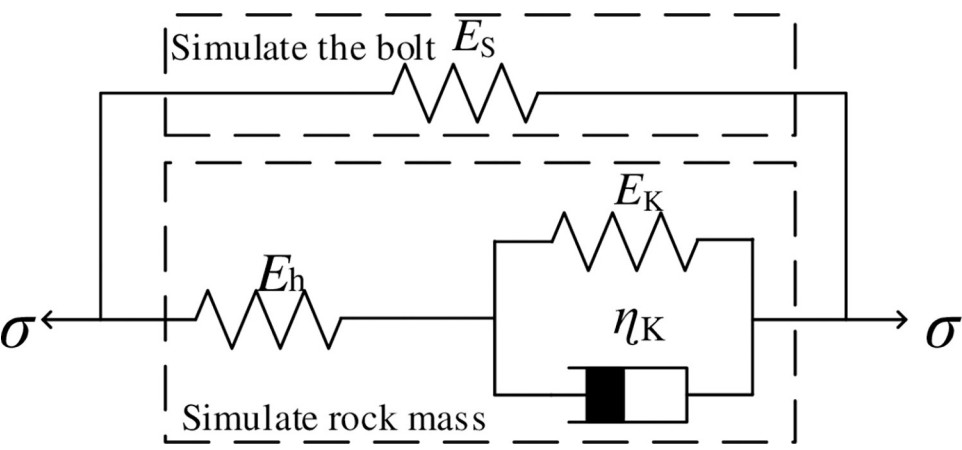

**Fig 10. Coupling model of the anchorage rock mass.**

When $\sigma = \sigma_c = const$, Eq (5) is transformed into the creep equation of the coupled model of the anchorage rock mass:

$$\varepsilon = e^{-\frac{M}{N}t} \times (C + \frac{\sigma_c}{M} e^{\frac{M}{N}t}) \tag{6}$$

where $M = \frac{E_h E_k + E_h E_s + E_s E_k}{E_h + E_k}$, $N = \frac{\eta_k (E_h + E_s)}{E_h + E_k}$, and C is the model parameter.

When $t = 0$, the instantaneous elastic deformation of the coupled model of the anchoring rock mass is $\varepsilon_0 = \sigma_c/(E_h + E_s)$, and $C = [\sigma_c/(E_h + E_s) - \sigma_c/M]$ is obtained after substituting into Eq (6). Then, Eq (6) can be translated into:

$$\varepsilon = e^{-\frac{M}{N}t} \times (\frac{\sigma_c}{E_h + E_s} - \frac{\sigma_c}{M} + \frac{\sigma_c}{M} e^{\frac{M}{N}t}) \tag{7}$$

## 5.2 Nonlinear rheologic element

The main reason for the nonlinear shear deformation of the anchorage rock mass is that the microcracks existing in the specimen material will undergo a nonlinear evolution and expansion fracture process with increasing time under the action of a constant shear stress level. Experts believe that the nonlinear shear creep deformation of anchored rock is a function of time, and it is assumed that the nonlinear shear creep deformation of anchored rock is a distribution function of time [1, 31].

The coupling effect of water on the anchorage rock mass leads to the accumulation of shear load in the attenuation and stability stages of the rock mass. When the crack penetrates and develops in the specimen in the steady state stage, the accumulated shear load is released in the acceleration stage so that the creep rate of the rock mass in the acceleration stage is faster than that of the rock mass in the anhydrous environment [32]. Therefore, the author introduced the change relationship between the shear modulus and water content in the anchorage rock mass into the traditional Weibull distribution function for correction so that it could meet the nonlinear change of shear creep of the anchorage rock mass under the coupling of water and rock. The corresponding nonlinear rheological element is shown in Fig 11. The creep equation of the nonlinear element is:

$$\varepsilon = \frac{\sigma_0}{G(\omega)} \{1 - m \times \exp[-(\frac{t - t_s}{t_p - t_s})^n]\} \tag{8}$$

where $\sigma_0$ is the initial stress, $G(\omega)$ is the change function of the shear modulus with water content, $m$ and $n$ are rheological parameters, $t_s$ is the transition time of the rock mass from the steady state stage to the accelerating stage, and $t_p$ is the shear failure time of the rock mass. It is worth noting that since Eq (2) considers the nonlinear change in the shear creep of rock, that is, the equation is valid under $t > t_s$, the nonlinear rheological element starts under this condition. Because the nonlinear change in rock is considered, the nonlinear rheological element can be used to study the acceleration stage of the anchoring rock mass under different water contents.

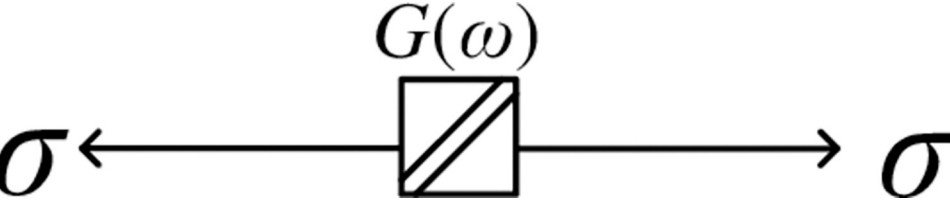

**Fig 11. Nonlinear rheological element.**

### 5.3 Water rock coupling model

The constitutive model of an anchoring rock mass can well describe the decay and stable creep deformation of rock [33]. To study the whole process of shear rheology of anchoring rock mass under different water content, nonlinear rheological elements were connected in series with the coupling model of anchoring rock mass to obtain the coupling model of water–rock, as shown in Fig 12. This model takes into account the deformation of rock bolts and the coupling effect of water on rock and can describe the whole process of shear creep of rock masses under water-bearing conditions. A combined water–rock coupling model based on the General Kelvin model, elastic body, and nonlinear rheological element, describing the creep responses of rock, anchor, and accelerated creep, respectively, was established. In addition, the coupling model has good applicability in describing the whole process of shear rheology of anchoring rock mass under different water content and provides a new framework for the stability of anchoring rock masses.

When $t \leq t_s$, the model degenerates into the coupled model of the anchorage rock mass, which can describe the attenuation and steady state stages of the anchorage rock mass under shear load:

$$\varepsilon = e^{-\frac{E_h E_k + E_h E_s + E_s E_k}{\eta_k (E_h + E_s)} t} \times \left[ \frac{\sigma_c}{E_h + E_s} - \frac{\sigma_c (E_h + E_k)}{E_h E_k + E_h E_s + E_s E_k} \left( 1 - e^{\frac{E_h E_k + E_h E_s + E_s E_k}{\eta_k (E_h + E_s)} t} \right) \right] \tag{9}$$

When $t > t_s$, the model is a water–rock coupling model, which can describe the whole process of attenuation, stabilization and acceleration of the anchorage rock mass under shear load:

$$\varepsilon = e^{-\frac{E_h E_k + E_h E_s + E_s E_k}{\eta_k (E_h + E_s)} t} \times \left[ \frac{\sigma_c}{E_h + E_s} - \frac{\sigma_c (E_h + E_k)}{E_h E_k + E_h E_s + E_s E_k} \left( 1 - e^{\frac{E_h E_k + E_h E_s + E_s E_k}{\eta_k (E_h + E_s)} t} \right) \right]$$
$$+ \frac{\sigma_0}{G(\omega)} \left\{ 1 - m e^{\left[ -\left( \frac{t - t_s}{t_p - t_s} \right)^n \right]} \right\} \tag{10}$$

### 6 Parameter solving and verification

Based on the quasi-Newton method and general global optimization method [34], the coupling model parameters were analysed by mathematical optimization software 1stOpt for the shear creep test curve. By fitting the test data and coupling the model equation to solve the parameters [35], relevant parameters are obtained, as shown in Table 3. As seen from Table 3, the fitting coefficients R2 are all above 0.97, indicating that the fitting effect of the test data and coupling model equation is good. It also shows that the coupled water–rock model considering

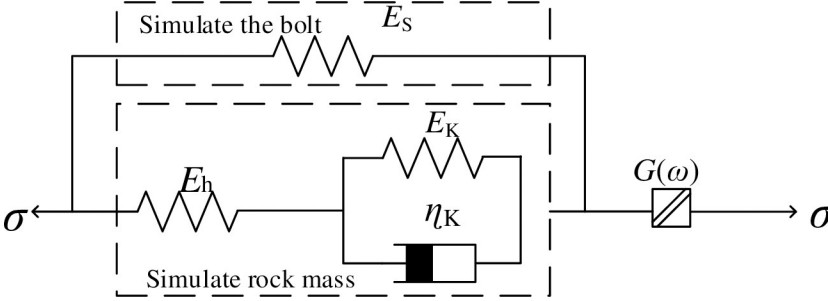

**Fig 12. Water–rock coupling model.**

**Table 3. Model parameters.**

| Moisture content/% | Shear stress/MPa | Rheological parameter | | Model coefficient | | | | $R^2$ |
|---|---|---|---|---|---|---|---|---|
| | | $m$ | $n$ | $E_k$/ GPa | $E_h$/ GPa | $E_s$/ GPa | $\eta_k$(GPa·h) | |
| 0 | 2 | | | 4.33 | 0.02 | 6.48 | 9.86 | 0.9738 |
| | 4 | | | 6.04 | 0.11 | 2.56 | 3.62 | 0.9861 |
| | 6 | | | 7.51 | 0.09 | 6.31 | 8.37 | 0.9931 |
| | 8 | 6.77 | 1.94 | 7.37 | 0.23 | 7.95 | 4.22 | 0.9881 |
| 0.08 | 2 | | | 4.69 | 0.01 | 2.19 | 7.83 | 0.9822 |
| | 4 | | | 2.49 | 0.03 | 3.36 | 5.37 | 0.9901 |
| | 6 | | | 3.82 | 0.35 | 2.33 | 7.95 | 0.9917 |
| | 8 | 9.51 | 4.92 | 5.56 | 0.76 | 3.74 | 4.81 | 0.9873 |
| 0.13 | 2 | | | 6.19 | 0.27 | 7.95 | 1.39 | 0.9840 |
| | 4 | | | 4.95 | 0.47 | 2.4 | 5.43 | 0.9946 |
| | 6 | | | 4.09 | 0.39 | 1.63 | 8.28 | 0.9928 |
| | 8 | 11.30 | 1.79 | 2.99 | 0.88 | 7.18 | 5.71 | 0.9844 |
| 0.16 | 2 | | | 8.26 | 0.55 | 4.65 | 3.75 | 0.9846 |
| | 4 | | | 6.98 | 0.61 | 4.38 | 7.98 | 0.9914 |
| | 6 | | | 1.89 | 0.37 | 2.49 | 5.79 | 0.9900 |
| | 8 | 7.81 | 1.99 | 3.88 | 0.33 | 4.78 | 9.53 | 0.9863 |

different water contents can better reflect the creep characteristics of the anchorage rock mass under shear loading.

To quantitatively illustrate the changing relationship between the coupled model and the test data, the comparison graphs of different models and shear creep curves were obtained through calculation, as shown in Fig 13. It can be seen from the figure that the Nishihara model can well describe the shear creep of the anchorage rock mass in the decay stage, but the description results of the unstable stage and acceleration stage are relatively poor, and the linear characteristics of the Nishihara model are obvious [36–38]. The coupled water–rock model with nonlinear rheological elements can not only accurately reflect the variation law of the specimen in the decay stage but also well explain the shear creep deformation of the anchorage rock mass in the unstable stage and the acceleration stage [39, 40]. Therefore, the coupled water–rock model considering water content can accurately analyse and study the whole process of shear creep of an anchorage rock mass.

By connecting the nonlinear rheological element with the coupled model of anchoring rock mass in series, the coupled model of water–rock under water cut conditions can be obtained, which can be used to study and analyse the steady creep of the anchorage rock mass under different water contents. The fitting curve of the stable stage is shown in Fig 14. It can be seen from the figure that the coupled model can well reflect the change law of the anchorage rock mass in the stable stage. The proposed model provides a theoretical basis for the study of anchored rock masses under water cut conditions and new methods for the stability and reinforcement of rock masses.

## 7 Conclusion

In this paper, marble is selected as the bedrock to carry out shear creep tests of anchoring rock masses under different water contents. Through the investigation of shear creep tests and constitutive models, the conclusions are as follows:

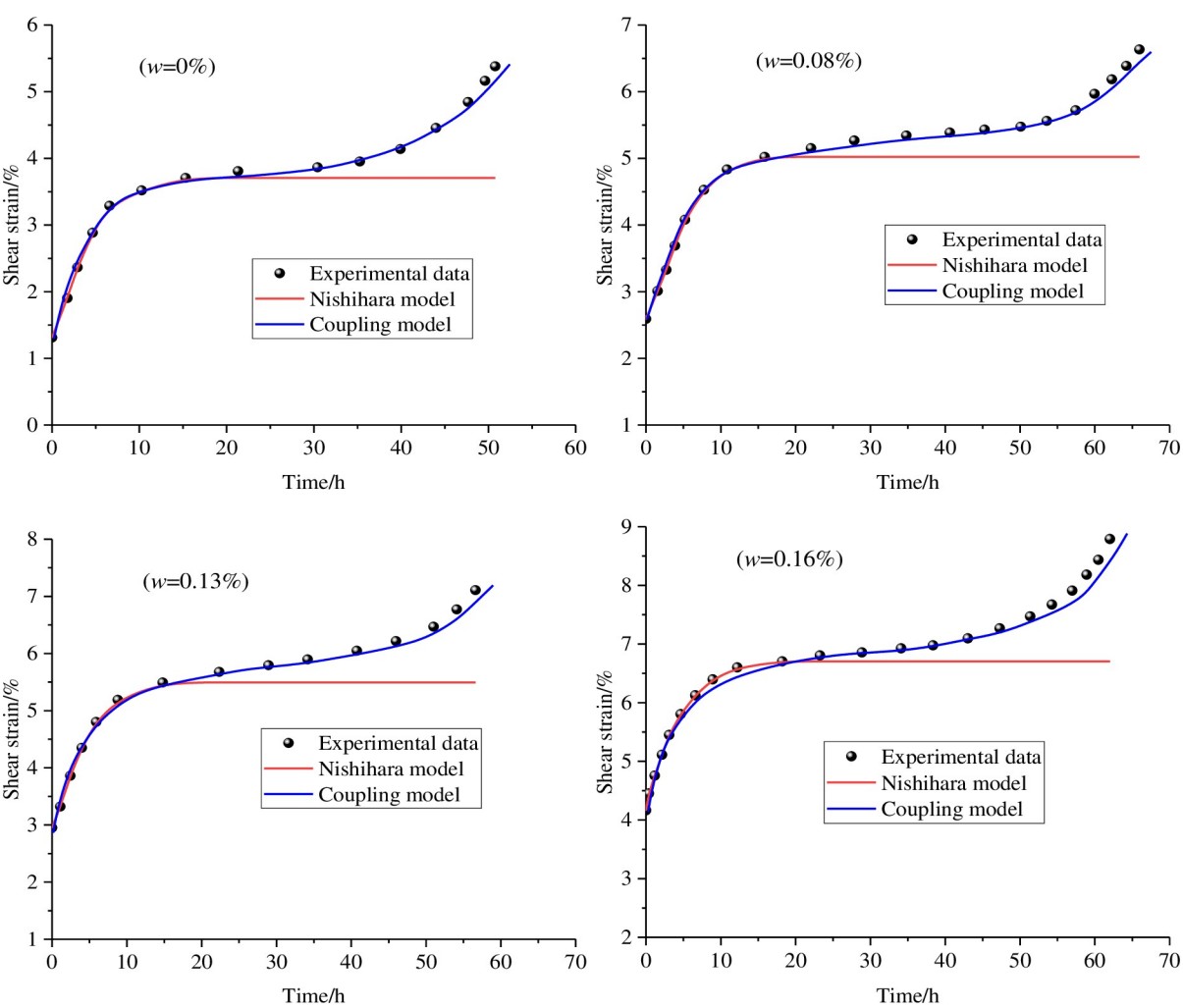

**Fig 13. Comparison curve between test data and coupling model.**

1. The creep variable of the anchorage rock mass at the steady state stage increases gradually with increasing water content under different water content conditions, and the deformation of the specimen with a water content of 0.16% is approximately 1.4 times that under dry conditions. The anchorage rock mass has typical decay, stability and acceleration stages under shear loading.

2. The long-term strength of the anchorage rock mass gradually decreases with increasing water content, mainly for the following reasons: the water content under shear loading plays a role in weakening the shear strength of the anchorage rock mass. The curve of the creep rate of the anchored rock mass first decreases rapidly and then tends to stabilize gradually, and the curve shows a U-shaped change under a high stress level. The increase in moisture content will promote the increase in the creep rate of the specimen.

3. The generalized Kelvin model can be connected in series with elastic elements to describe the creep characteristics of the anchorage rock mass in the decay and stability stages. The nonlinear rheological element describing the relationship between the water content and rock strength is introduced into the coupled model of the anchorage rock mass, and the

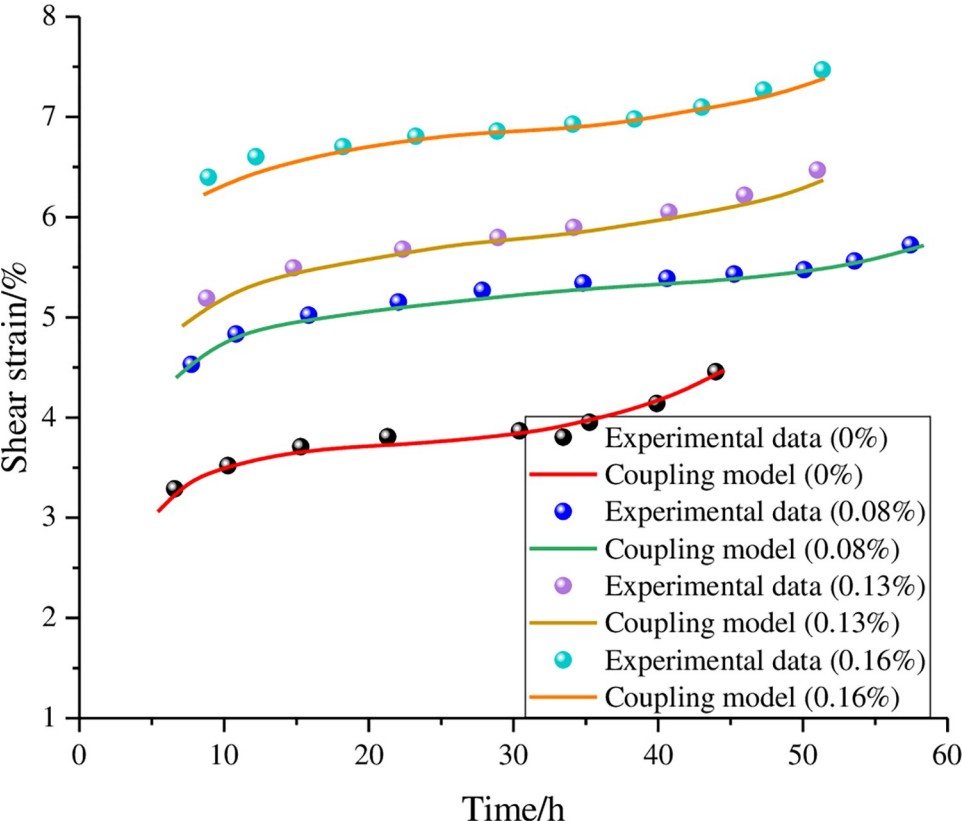

**Fig 14. Steady-state phase fitting curve.**

corresponding coupled model of rock water can be obtained. The comparison between experimental data and theoretical curves shows that the water–rock coupling model can well reflect the whole process of shear creep of an anchorage rock mass in a water-bearing environment.

## Author Contributions

**Conceptualization:** Zhao Zhongliang, Dai Wukui.

**Data curation:** Zhou Mi.

**Formal analysis:** Zhao Zhongliang, Dai Wukui.

**Investigation:** Yang Jianjun.

**Methodology:** Yang Jianjun, Zhou Mi.

**Project administration:** Dai Wukui.

**Validation:** Liu Ziwei.

**Visualization:** Liu Ziwei.

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
