## [Decision Letter · Decision Letter 0]

28 Nov 2022

PONE-D-22-28363Shear rheological model of the anchorage rock mass under water‒rock couplingPLOS ONE

Dear Dr. yang,

Thank you for submitting your manuscript to PLOS ONE. After careful consideration, we feel that it has merit but does not fully meet PLOS ONE’s publication criteria as it currently stands. Therefore, we invite you to submit a revised version of the manuscript that addresses the points raised during the review process. Please, address all the comments made by the reviewers.

We look forward to receiving your revised manuscript.

Kind regards,

Antonio Riveiro Rodríguez, PhD

Academic Editor

PLOS ONE

Journal Requirements:

"Author Contributions: Conceptualization, Z.Z.L and D.W.; methodology, Y.J.J.; software, Z.M.; validation, L.Z.W.; formal analysis, Z.Z.L and D.W.; investigation, Y.J.J.; resources, Y.J.J.; data curation, Y.J.J.; writing—original draft preparation, Z.Z.L; writing—review and editing, Z.Z.L; visualization, Z.Z.L; supervision, Z.Z.L; project administration, D.W.; funding acquisition, Z.Z.L. All authors have read and agreed to the published version of the manuscript.

Funding: This research was funded by the National Natural Science Foundation of China, grant number 51774166; Cscec Technology R&D Project, grant number CSCEC-2020-Z-57."

4. Thank you for stating the following in the Funding Section of your manuscript: 

"This research was funded by the National Natural Science Foundation of China, grant number 51774166; Cscec Technology R&D Project, grant number CSCEC-2020-Z-57."

"Author Contributions: Conceptualization, Z.Z.L and D.W.; methodology, Y.J.J.; software, Z.M.; validation, L.Z.W.; formal analysis, Z.Z.L and D.W.; investigation, Y.J.J.; resources, Y.J.J.; data curation, Y.J.J.; writing—original draft preparation, Z.Z.L; writing—review and editing, Z.Z.L; visualization, Z.Z.L; supervision, Z.Z.L; project administration, D.W.; funding acquisition, Z.Z.L. All authors have read and agreed to the published version of the manuscript.

Funding: This research was funded by the National Natural Science Foundation of China, grant number 51774166; Cscec Technology R&D Project, grant number CSCEC-2020-Z-57."

Reviewers' comments:

Reviewer's Responses to Questions

**Comments to the Author**

1. Is the manuscript technically sound, and do the data support the conclusions?

Reviewer #1: Yes

Reviewer #2: Yes

2. Has the statistical analysis been performed appropriately and rigorously? 

Reviewer #1: Yes

Reviewer #2: No

3. Have the authors made all data underlying the findings in their manuscript fully available?

Reviewer #1: Yes

Reviewer #2: Yes

4. Is the manuscript presented in an intelligible fashion and written in standard English?

Reviewer #1: Yes

Reviewer #2: Yes

5. Review Comments to the Author

Reviewer #1: In this paper, to study the shear creep deformation law of the anchoring rock mass under different water content conditions, shear creep tests of the anchoring rock mass under different water contents were carried out and the influence of water content on rock rheological haracteristics is explored by analysing the related mechanical properties of the anchorage rock mass. Then, the coupling model of the anchorage rock mass can be obtained by connecting the nonlinear rheological element and the coupling model of the anchorage rock mass in series. The paper is interesting and well written. But it still has some issues to be explained by the author before meeting the publishing standard.

(1)Please further explain the applicability of the model?

(2)The innovation of this paper is suggested to improve and highlight.

(3)What’s the “WEBER distribution? is the “Weibull distribution?

(4)In Figure 5, the cuves of 0.08% and 0.13% are same, please check. And the analysis of this section is not clear.

(5)In section 6, What's the role of this literature? I don't find the relationship between the literature content and this content. So PLE specify how to obtain the data in Table 2 to ensure the credibility of the data.

(6)Please check the information correction of this literature.

Reviewer #2: Water is an important factor to determine the mechanical behaviors of rock in the field condition. To study the shear creep deformation law of the anchoring rock mass under different water content conditions. This manuscript presented a Shear rheological model of the anchorage rock mass under water‒rock coupling. This manuscript is interesting. However, after a careful review of the paper, lots of details need to be revised. Therefore, it is recommended to revise the paper to increase its readability. Here are some suggestions.

1. The main content in this manuscript contains the creep mechanical tests and the sheal rheological model. The title does not cover the creep mechanical tests described in this manuscript. Thus, it is recommended to modify the title to contain the whole contents.

2. The first paragraph in Introduction, a key sentence is lacked to describe the important of the study work in this manuscript.

3. The second sentence in the second paragraph in Introduction has no meaning, please delete it.

4. “Zhao Tongbin et al.” should be “Zhao et al.”

5. The last paragraph does not need to state the results obtained in this study, the results should be summarized in the Conclusion. It should just describe the work you would like to do.

6. In Figure 1, the size of the anchor is suggested to be labeled.

7. The scheme of the conventional mechanical tests to obtain the parameters in Table 1 should be describe in detail. Did the uniaxial compression tests and direct shear tests used in this study? For the direct shear tests, the different normal stresses should be given, in this case, the cohesion and internal friction angle can be obtained. As well as, the shear strength may be various for the different normal stresses.

8. A water absorption test should be given and the curve of the water content changed with time must be given to demonstrate the relationship of the average water content and time as presented in Section 2.1.

9. The compositions and functions of the test equipment is recommended to added in Section 2.2.

10. A section to describe the testing scheme should be given in Section 2 to state the detail parameters setting for the different shear creep tests.

11. Section 2.3 is recommended to move into Section 3, and the title of Section 3 is suggested to be “results and analysis”

12. The description of the creep characteristics presented in Section 2.3 is unseemly. The creep stages of the specimen should be discussed according to the stress and time, some of them have three stages (under high stress), but some of them just have two stages (under low stress).

13. In Section 3.1, a curve to describe the relationship of water content and long-term strength should be given.

14. The value of the long-term strength should be given in Section 3.1.

15. Some part in this manuscript described unclearly, for example, “The curve is approximately linear before the long-term strength”, which curve should be given; “When the shear stress is greater than the long-term strength, the curve is approximately horizontal”, which curve is horizontal, but I cannot find a horizontal curve.

16. Can this shear rheological model describe the steady creep of the anchorage rock mass under water‒rock coupling? If it can, please demonstrate it.

6. PLOS authors have the option to publish the peer review history of their article (what does this mean?). If published, this will include your full peer review and any attached files.

Reviewer #1: No

Reviewer #2: No

---

## [Author Response · Author response to Decision Letter 0]

11 Feb 2023

Response to the comments of manuscript RMRE-D-22-00429R1

We thank the editor and reviewers for the time and effort that they put in to carefully review our manuscript. The comments were very helpful for improving the manuscript. We have carefully modified the corresponding contents, which are in red font in the revised manuscript.

The responses to each comment are listed as follows:

Associate Editor:

Thank you for submitting your manuscript to PLOS ONE. After careful consideration, we feel that it has merit but does not fully meet PLOS ONE’s publication criteria as it currently stands. Therefore, we invite you to submit a revised version of the manuscript that addresses the points raised during the review process.

Reply: Thank you for the suggestions. We have revised the paper based on each comment.

Reviewer #1:

In this paper, to study the shear creep deformation law of the anchoring rock mass under different water content conditions, shear creep tests of the anchoring rock mass under different water contents were carried out and the influence of water content on rock rheological haracteristics is explored by analysing the related mechanical properties of the anchorage rock mass. Then, the coupling model of the anchorage rock mass can be obtained by connecting the nonlinear rheological element and the coupling model of the anchorage rock mass in series. The paper is interesting and well written. But it still has some issues to be explained by the author before meeting the publishing standard.

(1). Please further explain the applicability of the model?

Reply: Thank you for the suggestions. We have supplemented the applicability of the coupling model in the revised manuscript as follows:

A combined water‒rock coupling model based on the General Kelvin model, elastic body, and nonlinear rheological element, describing the creep responses of rock, anchor, and accelerated creep, respectively, was established. In addition, the coupling model has good applicability in describing the whole process of shear rheology of anchoring rock mass under different water content and provides a new framework for the stability of anchoring rock masses.

(2). The innovation of this paper is suggested to improve and highlight.

Reply: Thank you for the suggestions. In this paper, the timeliness of anchorage rock mass under water-rock coupling environment is studied and related tests are carried out. In this paper, the original water-rock coupling model was optimized by exploring the quantitative relationship between water content and shear modulus, and nonlinear rheological elements are introduced to characterize the deformation characteristics of the anchorage rock mass at the acceleration stage. Relevant innovation points have been added in the paper.

(3). What’s the “WEBER distribution? is the “Weibull distribution?

Reply: Thank you for the suggestions. We have used "Weibull distribution" to replace "WEBER distribution" in the revised manuscript according to the expert's advice.

(4). In Figure 5, the cuves of 0.08% and 0.13% are same, please check. And the analysis of this section is not clear.

Reply: Thank you for the suggestions. The two pictures are the same because of our mistake. We have modified the pictures in the article. Meanwhile, we have supplemented and explained relevant contents in the revised manuscript as follows:

As can be seen from the figure, the curve cluster gradually diverges from aggregation, marking the transformation of the specimen from viscoelastic deformation stage to viscoplastic deformation stage, indicating that there is an obvious divergence starting point of the curve.

(5). In section 6, What's the role of this literature? I don't find the relationship between the literature content and this content. So PLE specify how to obtain the data in Table 2 to ensure the credibility of the data.

Reply: Thank you for the suggestions. In section 6, there are some problems in the use of relevant literatures. We have modified and adjusted the literatures to meet the requirements of the paper. The method for determining parameters in Table 2 is as follows:

The accuracy and effectiveness of the model can be verified by the fitting analysis combined with the experimental results. For nonlinear problems, BFGS algorithm and general global optimization method in mathematical analysis software 1stOpt are widely used to carry out nonlinear fitting of test data and calculate creep parameters through inversion.

(6). Please check the information correction of this literature.

Reply: Thank you for the suggestions. We have checked and adjusted the literatures in the paper.

Reviewer #2:

Water is an important factor to determine the mechanical behaviors of rock in the field condition. To study the shear creep deformation law of the anchoring rock mass under different water content conditions. This manuscript presented a Shear rheological model of the anchorage rock mass under water‒rock coupling. This manuscript is interesting. However, after a careful review of the paper, lots of details need to be revised. Therefore, it is recommended to revise the paper to increase its readability. Here are some suggestions.

(1). The main content in this manuscript contains the creep mechanical tests and the sheal rheological model. The title does not cover the creep mechanical tests described in this manuscript. Thus, it is recommended to modify the title to contain the whole contents. 

Reply: Thank you for the suggestions. We have revised the title of the paper according to the opinions of the reviewers.

(2). The first paragraph in Introduction, a key sentence is lacked to describe the important of the study work in this manuscript.

Reply: Thank you for the suggestions. We have added the important of this research to the first paragraph of the introduction. The relevant content is as follows:

Different from those encountered during shallow geotechnical engineering, the geological environments of deep rocks are extremely complex and can be summarized by a high temperature, high stress, high permeability and underground water, among which a large underground water is one of the most common triggers of geological disasters. The study of creep mechanical tests and shear rheological model of the anchorage rock mass under water‒rock coupling will facilitate a better understanding of the long-term stability of rock masses.

(3). The second sentence in the second paragraph in Introduction has no meaning, please delete it.

Reply: Thank you for the suggestions. We have deleted the second sentence in the second paragraph in Introduction as requested.

(4). “Zhao Tongbin et al.” should be “Zhao et al.”

Reply: Thank you for the suggestions. We have modified the reference and adjusted the same problems in the paper.

(5). The last paragraph does not need to state the results obtained in this study, the results should be summarized in the Conclusion. It should just describe the work you would like to do.

Reply: Thank you for the suggestions. We have revised and adjusted the relevant content in this paper according to the opinions of experts.

(6). In Figure 1, the size of the anchor is suggested to be labeled.

Reply: Thank you for the suggestions. We have marked the size of anchor in Figure 1 in the paper.

(7). The scheme of the conventional mechanical tests to obtain the parameters in Table 1 should be describe in detail. Did the uniaxial compression tests and direct shear tests used in this study? For the direct shear tests, the different normal stresses should be given, in this case, the cohesion and internal friction angle can be obtained. As well as, the shear strength may be various for the different normal stresses.

Reply: Thank you for the suggestions. The mechanical properties of the specimen were obtained by uniaxial compression test, tensile test and direct shear test respectively. We supplemented the source of mechanical properties of tested samples in the revised manuscript as follows:

A uniaxial compression test was used to determine the uniaxial compressive strength (UCS), Elastic modulus and Poisson's ratio of the specimen, while shear strength, cohesion and internal friction angle were obtained by a direct shear test. For the mechanical parameters of the specimen, the UCS was tested in the direction parallel to the joint, the direct shear test was conducted along the joint.

The relevant test results are shown in Fig. 1-2.

Fig. 1 Uniaxial stress-strain curve

Fig. 2 Peak shear stress curve under normal stress

(8). A water absorption test should be given and the curve of the water content changed with time must be given to demonstrate the relationship of the average water content and time as presented in Section 2.1.

Reply: Thank you for the suggestions. We have supplemented the water content test and corresponding test results in the paper, and the relevant contents are as follows:

Fig. 3 Water content test: (a) immersion device; (b) test results

(9). The compositions and functions of the test equipment is recommended to added in Section 2.2.

Reply: Thank you for the suggestions. We have added the composition and function of the test equipment in Section 2.2, as shown below:

The TAW2000 testing machine is composed of a loading system, measuring system, controller and other parts. It adopts microcomputer-controlled electrohydraulic servo valve loading and manual hydraulic loading to complete automatic control. The testing machine have the ability to automatically complete the rocks’ uniaxial and triaxial compression tests, uniaxial & triaxial rheological tests, and shear composite tests. In the test, the electro-hydraulic servo proportional valve group with wide range of speed regulation and computer digital control is applied to automatically and accurately realize the tests of the axial and radial constant stress, the constant strain, and the constant displacement. It can dynamically display the whole process of the test.

(10). A section to describe the testing scheme should be given in Section 2 to state the detail parameters setting for the different shear creep tests.

Reply: Thank you for the suggestions. We have added the testing scheme in the paper as follows:

Table 1 Test scheme

Test group Water content/% Shear stress/MPa Shear strength/MPa

Group A 0 2-4-6-8 10

Group B 0.08 2-4-6-8 10

Group C 0.13 2-4-6-8 10

Group D 0.16 2-4-6-8 10

(11). Section 2.3 is recommended to move into Section 3, and the title of Section 3 is suggested to be “results and analysis”

Reply: Thank you for the suggestions. We have adjusted the chapters of the manuscript according to the requirements of the reviewers.

(12). The description of the creep characteristics presented in Section 2.3 is unseemly. The creep stages of the specimen should be discussed according to the stress and time, some of them have three stages (under high stress), but some of them just have two stages (under low stress).

Reply: Thank you for the suggestions. We have revised it in the article. It can be seen from the test curve that the creep characteristics under different stress levels are different. The creep deformation under high stress includes decay, steady and acceleration stages, while the creep deformation under low stress only includes decay and stability stages.

(13). In Section 3.1, a curve to describe the relationship of water content and long-term strength should be given.

Reply: Thank you for the suggestions. We have added water content and long-term strength curve to the paper according to the opinions of reviewers as shown below:

Fig. 4 Curve of long-term strength with water content

(14). The value of the long-term strength should be given in Section 3.1.

Reply: Thank you for the suggestions. We have labeled and added the long-term strength values in Figure 5.

(15). Some part in this manuscript described unclearly, for example, “The curve is approximately linear before the long-term strength”, which curve should be given; “When the shear stress is greater than the long-term strength, the curve is approximately horizontal”, which curve is horizontal, but I cannot find a horizontal curve.

Reply: Thank you for the suggestions. We have marked and explained the linear curve in this paper. The description of the horizontal curve is inappropriate, and we have modified it.

(16). Can this shear rheological model describe the steady creep of the anchorage rock mass under water‒rock coupling? If it can, please demonstrate it.

Reply: Thank you for the suggestions. The rheological model can describe the steady-state creep of the anchorage rock mass under water-rock coupling as follows:

Fig. 5 Steady-state phase fitting curve

---

## [Decision Letter · Decision Letter 1]

7 Mar 2023

PONE-D-22-28363R1Creep mechanical tests and shear rheological model of the anchorage rock mass under water‒rock couplingPLOS ONE

Dear Dr. yang,

Thank you for submitting your manuscript to PLOS ONE. After careful consideration, we feel that it has merit but does not fully meet PLOS ONE’s publication criteria as it currently stands. Therefore, we invite you to submit a revised version of the manuscript that addresses the points raised during the review process.

Please, address all the points indicated by reviewer 2. Please submit your revised manuscript by Apr 21 2023 11:59PM. If you will need more time than this to complete your revisions, please reply to this message or contact the journal office at plosone@plos.org. Please include the following items when submitting your revised manuscript:A rebuttal letter that responds to each point raised by the academic editor and reviewer(s). You should upload this letter as a separate file labeled 'Response to Reviewers'.A marked-up copy of your manuscript that highlights changes made to the original version. You should upload this as a separate file labeled 'Revised Manuscript with Track Changes'.An unmarked version of your revised paper without tracked changes. You should upload this as a separate file labeled 'Manuscript'.If applicable, we recommend that you deposit your laboratory protocols in protocols.io to enhance the reproducibility of your results. Protocols.io assigns your protocol its own identifier (DOI) so that it can be cited independently in the future. For instructions see: https://journals.plos.org/plosone/s/submission-guidelines#loc-laboratory-protocols. Additionally, PLOS ONE offers an option for publishing peer-reviewed Lab Protocol articles, which describe protocols hosted on protocols.io. Read more information on sharing protocols at https://plos.org/protocols?utm_medium=editorial-email&utm_source=authorletters&utm_campaign=protocols.

We look forward to receiving your revised manuscript.

Kind regards,

Antonio Riveiro Rodríguez, PhD

Academic Editor

PLOS ONE

Journal Requirements:

Reviewers' comments:

Reviewer's Responses to Questions

**Comments to the Author**

1. If the authors have adequately addressed your comments raised in a previous round of review and you feel that this manuscript is now acceptable for publication, you may indicate that here to bypass the “Comments to the Author” section, enter your conflict of interest statement in the “Confidential to Editor” section, and submit your "Accept" recommendation.

Reviewer #1: All comments have been addressed

Reviewer #2: (No Response)

2. Is the manuscript technically sound, and do the data support the conclusions?

Reviewer #1: Yes

Reviewer #2: (No Response)

3. Has the statistical analysis been performed appropriately and rigorously? 

Reviewer #1: Yes

Reviewer #2: (No Response)

4. Have the authors made all data underlying the findings in their manuscript fully available?

Reviewer #1: (No Response)

Reviewer #2: (No Response)

5. Is the manuscript presented in an intelligible fashion and written in standard English?

Reviewer #1: (No Response)

Reviewer #2: (No Response)

6. Review Comments to the Author

Reviewer #1: (No Response)

Reviewer #2: 1.Authors gave the results of uniaxial compression tests and direct shear tests as shown in the Response to the comments, but they are not found in the manuscript, please add them in the manuscript.

2.For the problem of 16 that I stated before: “Can this shear rheological model describe the steady creep of the anchorage rock mass under water‒rock coupling? If it can, please demonstrate it.” The response just shows a figure, and it has no a word to describe it. As well as, this content should be incorporated into the manuscript.

7. PLOS authors have the option to publish the peer review history of their article (what does this mean?). If published, this will include your full peer review and any attached files.

Reviewer #1: No

Reviewer #2: **Yes: **Qiangui Zhang

---

## [Author Response · Author response to Decision Letter 1]

9 Mar 2023

Response to the comments of manuscript RMRE-D-22-00429R1

We thank the editor and reviewers for the time and effort that they put in to carefully review our manuscript. The comments were very helpful for improving the manuscript. We have carefully modified the corresponding contents, which are in red font in the revised manuscript.

The responses to each comment are listed as follows:

Associate Editor:

Thank you for submitting your manuscript to PLOS ONE. After careful consideration, we feel that it has merit but does not fully meet PLOS ONE’s publication criteria as it currently stands. Therefore, we invite you to submit a revised version of the manuscript that addresses the points raised during the review process.

Reply: Thank you for the suggestions. We have revised the paper based on each comment.

Reviewer #2:

(1). Authors gave the results of uniaxial compression tests and direct shear tests as shown in the Response to the comments, but they are not found in the manuscript, please add them in the manuscript.

Reply: Thank you for the suggestions. We have supplemented the figure in the revised manuscript as follows:

The relevant test figures are shown in Fig. 1-2.

Fig. 1 Uniaxial stress-strain curve

Fig. 2 Peak shear stress curve under normal stress

(2). For the problem of 16 that I stated before: “Can this shear rheological model describe the steady creep of the anchorage rock mass under water‒rock coupling? If it can, please demonstrate it.” The response just shows a figure, and it has no a word to describe it. As well as, this content should be incorporated into the manuscript.

Reply: Thank you for the suggestions. The rheological model can describe the steady-state creep of the anchorage rock mass under water-rock coupling. We have supplemented the figure and description in the revised manuscript as follows:

Fig. 3 Steady-state phase fitting curve

By connecting the nonlinear rheological element with the coupled model of anchoring rock mass in series, the coupled model of water‒rock under water cut conditions can be obtained, which can be used to study and analyse the steady creep of the anchorage rock mass under different water contents. The fitting curve of the stable stage is shown in Figure 14. It can be seen from the figure that the coupled model can well reflect the change law of the anchorage rock mass in the stable stage. The proposed model provides a theoretical basis for the study of anchored rock masses under water cut conditions and new methods for the stability and reinforcement of rock masses.

---

## [Decision Letter · Decision Letter 2]

4 Apr 2023

Creep mechanical tests and shear rheological model of the anchorage rock mass under water‒rock coupling

PONE-D-22-28363R2

Dear Dr. yang,

We’re pleased to inform you that your manuscript has been judged scientifically suitable for publication and will be formally accepted for publication once it meets all outstanding technical requirements.

Kind regards,

Antonio Riveiro Rodríguez, PhD

Academic Editor

PLOS ONE

Reviewers' comments:

Reviewer's Responses to Questions

**Comments to the Author**

1. If the authors have adequately addressed your comments raised in a previous round of review and you feel that this manuscript is now acceptable for publication, you may indicate that here to bypass the “Comments to the Author” section, enter your conflict of interest statement in the “Confidential to Editor” section, and submit your "Accept" recommendation.

Reviewer #2: All comments have been addressed

2. Is the manuscript technically sound, and do the data support the conclusions?

Reviewer #2: Yes

3. Has the statistical analysis been performed appropriately and rigorously? 

Reviewer #2: Yes

4. Have the authors made all data underlying the findings in their manuscript fully available?

Reviewer #2: Yes

5. Is the manuscript presented in an intelligible fashion and written in standard English?

Reviewer #2: Yes

6. Review Comments to the Author

Reviewer #2: (No Response)

7. PLOS authors have the option to publish the peer review history of their article (what does this mean?). If published, this will include your full peer review and any attached files.

Reviewer #2: **Yes: **Qiangui Zhang

---

## [Editor Report · Acceptance letter]

6 Apr 2023

PONE-D-22-28363R2 

Creep mechanical tests and shear rheological model of the anchorage rock mass under water‒rock coupling 

Dear Dr. Jianjun:

I'm pleased to inform you that your manuscript has been deemed suitable for publication in PLOS ONE. Congratulations! Your manuscript is now with our production department. 

Kind regards, 

on behalf of

Dr. Antonio Riveiro Rodríguez 

Academic Editor

PLOS ONE